# The Role of Bcl-2 and Beclin-1 Complex in “Switching” between Apoptosis and Autophagy in Human Glioma Cells upon LY294002 and Sorafenib Treatment

**DOI:** 10.3390/cells12232670

**Published:** 2023-11-21

**Authors:** Adrian Zając, Aleksandra Maciejczyk, Joanna Sumorek-Wiadro, Kamil Filipek, Kamil Deryło, Ewa Langner, Jarosław Pawelec, Magdalena Wasiak, Mateusz Ścibiorski, Wojciech Rzeski, Marek Tchórzewski, Michał Reichert, Joanna Jakubowicz-Gil

**Affiliations:** 1Department of Functional Anatomy and Cytobiology, Institute of Biological Sciences, Maria Curie-Skłodowska University, Akademicka 19, 20-033 Lublin, Poland; olamaciejczyk77@gmail.com (A.M.); joanna.sumorek-wiadro@mail.umcs.pl (J.S.-W.); mscibiorski1999@gmail.com (M.Ś.); wojciech.rzeski@mail.umcs.pl (W.R.); joanna.jakubowicz-gil@mail.umcs.pl (J.J.-G.); 2Department of Molecular Biology, Institute of Biological Sciences, Maria Curie-Skłodowska University, 20-033 Lublin, Poland; kamil.filipek2@mail.umcs.pl (K.F.); kamil@hektor.umcs.lublin.pl (K.D.); marek.tchorzewski@mail.umcs.pl (M.T.); 3Department of Medical Biology, Institute of Rural Health, Jaczewskiego 2, 20-950 Lublin, Poland; ewa.langner@gmail.com; 4Institute Microscopy Laboratory, Maria Curie-Skłodowska University, Akademicka 19, 20-033 Lublin, Poland; jaroslaw.pawelec@mail.umcs.pl; 5Department of Pathological Anatomy, National Veterinary Research Institute, 57 Partyzantów Avenue, 24-100 Puławy, Poland; magdalena.wasiak@piwet.pulawy.pl (M.W.); reichert@piwet.pulawy.pl (M.R.)

**Keywords:** human gliomas, anaplastic astrocytoma, glioblastoma multiforme, apoptosis, autophagy, Bcl-2:beclin-1 complex, PI3K inhibitor, Raf inhibitor

## Abstract

Background: Gliomas are the most malignant tumors of the central nervous system. One of the factors in their high drug resistance is avoiding programmed death (PCD) induction. This is related to the overexpression of intracellular survival pathways: PI3K-Akt/PKB-mTOR and Ras-Raf-MEK-ERK. Apoptosis and autophagy are co-existing processes due to the interactions between Bcl-2 and beclin-1 proteins. Their complex may be a molecular “toggle-switch” between PCD types. The aim of this research was to investigate the role of Bcl-2:beclin-1 complex in glioma cell elimination through the combined action of LY294002 and sorafenib. Methods: Drug cytotoxicity was estimated with an MTT test. The type of cell death was evaluated using variant microscopy techniques (fluorochrome staining, immunocytochemistry, and transmission electron microscopy), as well as the Bcl-2:beclin-1 complex formation and protein localization. Molecular analysis of PCD indicators was conducted through immunoblotting, immunoprecipitation, and ELISA testing. SiRNA was used to block Bcl-2 and beclin-1 expression. Results: The results showed the inhibitors used in simultaneous application resulted in Bcl-2:beclin-1 complex formation and apoptosis becoming dominant. This was accompanied by changes in the location of the tested proteins. Conclusions: “Switching” between apoptosis and autophagy using PI3K and Raf inhibitors with Bcl-2:beclin-1 complex formation opens new therapeutic perspectives against gliomas.

## 1. Introduction

Gliomas are the most malignant tumors of the central nervous system in adults and constitute an extremely important therapeutic problem affecting people around the world. Currently available treatment methods only improve quality of life and slightly extend it. The high resistance to treatment is based on the very intense proliferative, migration, and infiltration potential of cells [1]. The malignant nature of gliomas and their resistance to programmed death induction is closely related to the overexpression of intracellular survival signaling pathways. Examples of such pathways include PI3K-Akt/PKB-mTOR and Ras-Raf-MEK-ERK, which in normal cells control the proliferation and survival of cells or the elimination of defective cells through programmed cell death (PCD) [2,3]. Mechanisms of programmed death are co-existing processes in cells, as well as interdependent and mutually affecting each other. This is possible due to the molecular interactions and changes in the regulators of PCD-I and PCD-II. An example of those relations is Atg autophagy-related proteins, which can also change the mitochondrial membrane potential and, consequently, induce apoptosis. Inhibition of these specific Atgs by apoptosis-regulating proteins may in turn inhibit this process in favor of autophagy. Another example of a correlation between both types of death is the fusion of the anti-apoptotic protein Bcl-2 and a specific autophagy marker beclin-1 [4,5,6]. The most frequently observed types of PCD in glioblastoma cells are apoptosis (type I) and autophagy (type II), which are regulated at the molecular level by numerous proteins, including Bcl-2 (apoptosis) and beclin-1 (autophagy). Bcl-2 is a member of the Bcl-2 family of proteins, such as Bcl-xL and Mcl-1, and it is well-known as an anti-apoptotic molecular regulator. Beclin-1 is a crucial initiator of autophagy and recruits key proteins to a pre-autophagosomal structure and regulates the proper formation of the core complex consisting of beclin-1, Vps34, and Vps15. It also plays a key role as a determinate of whether cells undergo autophagy or apoptosis, which seems to be important in the light of recent reports that autophagy may lead to cancer cell survival. In normal physiological conditions in cells, Bcl-2 inhibits the activity of beclin-1 by combining with it, which allows maintenance of death homeostasis. In stress, these proteins dissociate, which leads to the induction of autophagy. The correlation between these proteins occurs through the beclin-1 Bcl-2 homology 3 (BH3) domain, which was shown to interact with anti-apoptotic Bcl-2 family members, as well as Bcl-2. This anti-apoptotic regulator has a hydrophobic groove in its structure, which is formed of BH1, BH2, and BH3 domains and affects through beclin-1′s BH3 (Figure 1A) [7].

The affinity of BH3 of beclin-1 to the hydrophobic groove of Bcl-2 is the basis of these protein interactions and prevents beclin-1 from assembling the pre-autophagosomal structure that leads to autophagy inhibition. The fact that beclin-1 has the ability to bind to other members of the Bcl-2 family of proteins suggests that the physiological significance and role of this interaction may depend on the protein which it binds to. The formation of the Bcl-2:beclin-1 complex also depends on the intracellular localization of the proteins. Endoplasmic reticulum Bcl-2 can inhibit autophagy and promote the apoptosis process, while the mitochondrial protein has an anti-apoptotic effect by releasing beclin-1, which in turn promotes the autophagy process. The created complex of those proteins may be some kind of molecular “toggle-switch” between apoptosis and autophagy. It has been observed that a significant increase in the level of apoptosis in glioma cells is closely correlated with the presence of a complex of these proteins, and when the proteins are in the free state in the cell, the dominant type of death is autophagy (Figure 1B). The possibility of using such a “switch” between the I and II type of programmed death in modern therapy seems greatly beneficial, especially in the case of autophagy, as a cancer cell survival mechanism.

Our previous studies [3] have shown that the increased activity of intracellular survival signaling pathways—PI3K-Akt/PKB-mTOR and Ras-Raf-MEK-ERK—is responsible for the resistance of glioma cells to their elimination by apoptosis. Moreover, we observed that the usage of these pathways’ inhibitors, LY294002 (PI3K inhibitor) and sorafenib (Raf inhibitor), respectively, in a single application mainly induced autophagy. After the combined action of both compounds, mainly apoptosis was observed in the tested cells.

Therefore, the aim of this study was to assess, for the first time, the contribution of the formation of the Bcl-2 and beclin-1 complex in the context of the glioma cell elimination through programmed cell death induction using the combined action of LY294002 and sorafenib.

## 2. Materials and Methods

### 2.1. DNA Constructs

A coding sequence for Bcl-2 was introduced into GFP vector (AddGene, sequence map available in Appendix A). pEGFP-beclin-1 construct was a kind gift from Cambridge University (Acknowledgment) and was introduced into mCherry vector. All genetic constructs were verified through DNA sequencing. Sequences of PCR primers used to prepare genetic constructs are available in the Appendix A. For further analyses, EmGFP-Bcl-2 and pmCherryN-beclin-1 were used.

### 2.2. Cell and Culture Conditions and Co-Transfection

Four cell lines were used in the experiments: two were normal and included oligodendrocytes and astrocytes, and two were cancer cells. The Department of Neonatology, Charite, Campus Virchow Klinikum, Humboldt University, Berlin provided a permanent rat oligodendrocyte cell line (OLN-93) that was cultured in a 1:1 combination of Dulbecco’s Modified Eagle Medium (DMEM) and Ham’s nutritional mixture F-12 (Sigma, St. Louis, MO, USA). Astrocyte basal medium (ABM) (Lonza; Basel, Switzerland) supplemented with SingleQuots supplements (Lonza; Basel, Switzerland) was used to cultivate normal human astrocytes (NHA), which were procured from Lonza (Basel, Switzerland; Catalog: CC-2565). The human anaplastic astrocytoma (MOGGCCM) and glioblastoma multiforme (T98G) cells were cultivated in a 3:1 combination of Dulbecco’s Modified Eagle Medium (DMEM) and Ham’s nutrient mixture F-12 (Sigma, St. Louis, MO, USA). The cells were obtained from the European Collection of Cell Cultures, ECACC, Porton Down, Salisbury, UK. Then, 10% fetal bovine serum (FBS; Sigma, St. Louis, MO, USA), 100 units/mL of penicillin (Sigma, St. Louis, MO, USA), and 100 µg/mL of streptomycin (Sigma, St. Louis, MO, USA) were added to each cell culture. The cultures were maintained at 37 °C in a 95% air and 5% CO_2_ humidified environment. Cells were transiently transfected with DNA constructs using Lipofectamine^TM^ 2000 Transfection Reagent (Thermo Fisher Scientific, Walthman, MA, USA) according to the manufacturer’s instructions. Control of Lipofectamine^TM^ and autofluorescence of untreated and treated cells was performed (Appendix A). After 24 h of transfection, the cells were used for further analysis.

### 2.3. Drug Treatment

LY294002 (Sigma, St. Louis, MO, USA) and sorafenib (Nexavar, BAY 43–9006) at the most effective concentrations in single as well as in simultaneous application and with the proper time of incubation were selected through a screening analysis from our previous research [2,3] and after additional MTT test evaluation. Both inhibitors were dissolved in DMSO to obtain 0.01%, its final concentration, which was the same in all variants, including controls and experiments. As controls, cells were incubated with 0.01% of DMSO only. Cancer as well as control cells were incubated with LY294002 (10 µM) and sorafenib (1 µM) in a single application and simultaneously for 24 h.

### 2.4. Evaluation of Cytotoxic Effect Using a MTT Test

Cytotoxic effect on normal as well as cancer cells was evaluated using an MTT test. Cells were grown in 96-well plates at 3 × 10^4^ density and incubated with LY294002 or sorafenib in different concentrations (5, 10, 20, and 30 µM in case of LY294002 and 0.25, 0.5, 0.75, 1, and 5 μM for sorafenib) for 24 h. Afterward, 15 µL of 3-(4,5-Dimethylthiazol-2-yl)-2,5-diphenyltetrazolium bromide (MTT) reagent (Merck SA, Darmstadt, Germany) dissolved in PBS buffer (5 mg/mL concentration) was added to each well for 3 h at 37 °C. After this time, wells were supplemented with 100 µL of lysis solution (10% SDS in 0.01 M HCl), then the cell cultures were incubated in normal growth conditions for 24 h. After incubation, all plates were tested at 570 nm with a 800 TS (BioTek, Santa Clara, CA, USA) microplate reader.

### 2.5. Microscopic Detection of Apoptosis, Autophagy, and Necrosis with Fluorochromes

Staining with Hoechst 33342 (Sigma, St. Louis, MO, USA) and propidium iodide (Sigma, St. Louis, MO, USA) was chosen, as previously described [2,3], for the identification of necrosis and apoptosis, and staining with acridine orange (AO) was chosen for autophagy, in order to detect typical acidic vesicular organelles (AVOs). Glioma and normal cells were cultured in 8-well Lab-Tek^TM^ slides with a growth chamber (Nunc^TM^, Thermo Fisher Scientific, Walthman, USA). A confocal microscope (Axiovert 200 M with scanning head LSM 5 PASCAL, Zeiss) was used to analyze the morphology of dead cells. Observed typical morphological changes served as the foundation for classifying cell death types. After Hoechst 33342 staining, cells exhibiting blue fluorescent nuclei (fragmented or/and with condensed chromatin) were considered apoptotic. Necrotic cells showed pink fluorescence nuclei when exposed to propidium iodide. Autophagy was assumed for typical AO-positive cells that showed granular distribution of AVOs in the cytoplasm. Under a microscope, at least 1000 randomly chosen cells were counted. Each experiment was carried out three times.

### 2.6. Transmission Electron Microscopy (TEM)

The treated cells were taken and fixed at 4 °C for 2 h in 4% glutaraldehyde in 100 mM cacodylate buffer and 2 h in 1% osmium tetroxide. After being dehydrated in a succession of alcohol, the cells were embedded in LR white resin. On a microtome RMC MT-XL (Tucson, AZ, USA), ultrathin slices were cut with a diamond knife, gathered on formvar-coated copper grids, and contrasted using uranyl acetate and Reynold’s liquid. Each experimental version was subjected to at least 100 cell examinations. A Zeiss Libra 120 transmission electron microscope (Carl Zeiss SMT AG, Oberkochen, Germany) was used to examine the samples.

### 2.7. Caspase Activity Assay

According to our prior study, a SensoLyte^®^AMC Caspase Substrate Sampler Kit (AnaSpec, Fremont, CA, USA) was used to evaluate the activity of caspases 7, 8, and 9 in treated cells. Using 96-well black microplates and an 800 TS microplate reader from BioTek, Santa Clara, California, USA, the fluorescence of 7-aminocoumarin (AMC) was examined at Ex/Em = 354/422 nm.

### 2.8. Indirect Immunofluorescence

Following the incubation period with LY294002 and/or sorafenib, the cancer cells underwent three PBS washes before being fixed for seven minutes (experimental testing determined the appropriate duration) in 3.7% paraformaldehyde (Sigma, St. Louis, MO, USA). The cells were thoroughly washed with PBS, treated for 10 min with 0.2% Triton X-100 (Sigma, St. Louis, MO, USA), and then thoroughly washed three more times with room-temperature PBS. Subsequently, cell cultures were incubated for 30 min at room temperature using UltraCruz^®^ Blocking Reagent (Santa Cruz Biotechnology, Dallas, TX, USA). Afterwards, the cells were treated for an additional night at 4 °C with diluted 1:100 anti-Bcl-2 rabbit and anti-beclin-1 mouse monoclonal antibodies (Santa Cruz Biotechnology, Dallas, TX, USA). The following day, secondary antibodies anti-rabbit labeled with AlexaFluor 488^TM^ and anti-mouse conjugated with AlexaFluor 594^TM^ (Santa Cruz Biotechnology, Dallas, TX, USA) were used for incubation with the cells, after they had been washed three times with PBS. Using a laser scanning confocal microscopy system LSM780 Zeiss (Oberkochen, Germany) with an excitation wavelength λ = 488 nm for AlexaFluor488^TM^ and λ = 594 nm for AlexaFluor594^TM^, the protein localization in the cells was examined. Three separate experiments were carried out. Each experimental variant involved the analysis of more than 100 cells. Co-localization was evaluated with Pearson correlation co-efficiency using ImageJ version 1.8.0 software (r = 0 random distribution, r = 1 co-localization).

### 2.9. Confocal Microscopy

An AxioObserverZ.1 inverted microscope, which was equipped with a PlanApochromat 63x/1.40 Oil DIC M27 objective, two PMT (PhotoMultiplayerTube) detectors, a 32-channel GaAsP spectral detector, and an environmental chamber to regulate the CO_2_ concentration, air temperature, and humidity was used to perform live-cell imaging of the treated cells. The system was manufactured by Zeiss (Oberkochen, Germany). Both transfected MOGGCCM and T98G cells were cultivated in 35 × 10 mm glass-bottom dishes (Greiner BioOne, Kremsmünster, Austria) at a density of 2 × 10^5^ and were observed following a 24 h treatment with LY294002 and/or sorafenib in several variations. A 488 nm laser line (2% of power) for EmGFP-Bcl-2 and a 561 nm laser line (2% of power) for pmCherryN-beclin-1, respectively, were used for excitation, and analysis of the expression level and subcellular localization of GFP- and mCherry-hybrid proteins was performed. The PMT detector operated at a range of 500–560 nm for GFP and 600–670 nm for mCherry, and 1 AU was selected as the pinhole diameter. In order to conduct comparative analyses of dox-dose responsiveness, the GFP channel’s gain was consistently adjusted to 790.

### 2.10. Co-Immunoprecipitation

Co-immunoprecipitation was carried out according to the manufacturer protocol (Santa Cruz Biotechnology, Dallas, TX, USA). Cancer cells were grown in 5 × 10^7^ in 25 cm^2^ Nunc™ cell culture treated flasks with filter caps (Thermo Fisher Scientific, Walthman, USA) for 24 h in 37 °C. Cell cultures were incubated with ice-cold Pierce™ IP lysis buffer (Thermo Fisher Scientific, Walthman, USA) for 10 min and then centrifugated for 10 min at 4 °C at 10,000× *g*. The obtained supernatant was transferred to new tubes, agarose-conjugated protein A was added, and it was incubated for 30 min at 4 °C, gently mixing the samples every 5 min. After this, cells were then centrifuged for 30 s at 1000× *g* at 4 °C, and the obtained supernatants were incubated with specific primary antibodies at a concentration of 2 μg/mL for 2 h at 4 °C. After incubation samples had been centrifugated at 1000× *g* for 30 s at 4 °C, the supernatant was discarded and immunoprecipitates were used for electrophoretic separation and immunoblotting for further analysis.

### 2.11. Immunoblotting

Treated cell extracts were prepared by lysing cells in hot buffer containing 125 mM Tris–HCl pH 6.8, 4% SDS, 10% glycerol, and 100 mM DTT). The Bradford method and an 800 TS (BioTek, Santa Clara, CA, USA) microplate reader were used for protein concentration measure. Then, 80 μg of each sample’s proteins was separated using 10% SDS-PAGE and electroblotted onto PVDF transfer membranes with 0.45 μm pores (Thermo Scientific, Rockford, IL, USA). Membranes were blocked with 5% low fat milk for 30 min and then incubated with primary antibodies: mouse monoclonal antibody anti-Bcl-2, anti-beclin-1 (Santa Cruz Biotechnology, concentration 0.5 μg/mL) at 4 °C overnight. The next day, after three washes with PBS supplemented with 0.05% Triton X-100 (Sigma), the membranes were incubated with secondary antibodies conjugated with alkaline phosphatase (AP) for 2 h at room temperature. Proteins were detected with NBT/BCIP Solution (ABCAM, Waltham, MA, USA). The results obtained were analyzed qualitatively on the basis of the band thickness, width, and color depth. The quantitative analysis of protein bands was performed using ImageLab Windows Program version 6.1.0 software (BIO-RAD, Hercules, CA, USA). The data were normalized relative to GAPDH (Santa Cruz Biotechnology, concentration 0.5 μg/mL). Three independent experiments were performed. Whole blot membranes are attached as Appendix A.

### 2.12. T98G and MOGGCCM Transfection with siRNA

After being cultured for 24 h at 37 °C in a CO_2_ incubator at a density of 2 × 10^5^, cells were treated in accordance with our earlier research [2,3]. The medium was aspirated following washing with a 3:1 DMEM/Ham’s F-12 mixture devoid of serum and antibiotics. Next, 2 μL of either anti-Bcl-2 or anti-beclin-1 siRNA (Santa Cruz Biotechnology, Dallas, TX, USA) combined with 2 μL of transfection reagent (also from Santa Cruz Biotechnology, Dallas, TX, USA) were used to transfect the cells. Cell cultures were supplemented with a medium containing 20% FBS and a double dose of antibiotics, after being incubated for five hours at 37 °C in a CO_2_ incubator. After that, an additional 18 h of incubation was carried out. The transfected cells were used for further research after the media was replaced with a new one. According to the manufacturer’s procedure, control siRNA (Fluorescein Conjugates-A; Santa Cruz Biotechnology, Dallas, TX, USA) was applied for transfection efficiency monitoring during fluorescence analysis.

### 2.13. Statistical Analysis

For statistical evaluation, GraphPad Prism 5 (GraphPad Software Inc., San Diego, CA, USA) was used to conduct a one-way ANOVA test, followed by Dunnett’s multiple comparison analysis. The significance threshold was set at *p* < 0.05 for data, shown as mean ± standard deviation (SD). Similarly to the previous study, Chou–Talalay’s test was used to validate the optimal dose selection of the substances under study [3] (Appendix A).

## 3. Results

### 3.1. Cytotoxic Effect of LY294002 and Sorafenib

The cytotoxicity of the inhibitors used was evaluated with MTT assay. Normal (NHA and OLN-93) as well as cancer (MOGGCCM and T98G) cells were incubated for 24 h with different concentrations of LY294002 (5, 10, 20, and 30 µM) or sorafenib (0.25, 0.5, 0.75, 1, and 5 μM) in a single application. MTT testing revealed that low doses (5 and 10 µM) of LY294002 did not have a significant effect on the NHA and OLN-93 cell viability (Figure 2E). Similar observations were noticed after incubation with sorafenib at 0.25, 0.5, and 1 μM concentrations (Figure 3E). The largest decrease in astrocyte and oligodendrocyte viability was observed after 20 and 30 µM of PI3K inhibitor (Figure 2E) and 5 μM of Raf inhibitor (Figure 3E). Cancer cells turned out to be more sensitive to elimination with LY294002 and sorafenib incubation compared to normal cells (Figure 2E and Figure 3E). However, MOGGCCM and T98G cells were more resistant to Raf inhibitor anticancer activity than PI3K inhibitor. The most effective concentration of LY294002 for anaplastic astrocytoma cell elimination was 20 μM, and in the case of glioblastoma multiforme, it was 30 μM (Figure 2E). Moreover, 5 μM of sorafenib was the most effective anticancer agent concentration toward MOGGCCM cells and 1 μM toward T98G cells (Figure 3E).

It is known that not only is cytotoxicity important in cancer cell elimination, but also the type of death, especially in the case of a harmful influence of necrosis and a tentative role for autophagy in carcinogenesis and progression; therefore, the microscopic analysis of death types was evaluated before determination of the most efficient drug concentration.

### 3.2. Microscopic Detection of Apoptosis, Autophagy, and Necrosis

#### 3.2.1. Fluorescent Staining with Fluorochromes

The cancer (MOGGCCM and T98G) and normal (NHA and OLN-93) cells were incubated with LY294002 (10 µM) or/and sorafenib (1 µM) for 24 h. The concentrations were selected from previous studies and due to results from the performed MTT test. After that, cells were stained with dyes typical for apoptosis, autophagy, and necrosis: Hoechst 33342, acridine orange (AO), and propidium iodide, respectively, and characteristic changes in cells were observed (Figure 4A,B,C,D). As shown in Figure 4C,D and Figure 5C–F, there was no significant apoptotic or autophagic effect on normal cells. Necrosis was not observed at a significant level after single and simultaneous inhibitor application. LY294002 was more effective for MOGGCCM cell elimination but mainly induced autophagy (over 40%), while apoptosis was the dominant type of death when it was combined with sorafenib (almost 20%), as presented in Figure 4A. In Figure 4B, we can see that T98G sorafenib and LY294002 were efficient in cell elimination. Single application of PI3K inhibitor mainly induced apoptosis (30%) but autophagy was also observed at a significant level (20%), whereas in T98G cells incubated with sorafenib only, autophagy was dominant (40%). The simultaneous application of both inhibitors in glioblastoma multiforme cells resulted in a significant increase in apoptosis (53%) and a decrease in autophagy (12%). In both cancer cell lines, a non-significant level of necrosis was observed.

#### 3.2.2. TEM Microscopy

Diverse types of cell death were definitively confirmed at the ultrastructure level using the characteristic morphological features observed under a transmission electron microscope (TEM), in control as well as treated cells. Control MOGGCCM and T98G cells (Figure 6A) were characterized by the typical and unchanged composition of their compartments and cytoplasm. In anaplastic astrocytoma and glioblastoma multiforme cells treated with a combination of sorafenib and LY294002, apoptotic bodies were observed (Figure 6B), whereas single application in both cell lines generated observable autophagic vesicular organelles (AVOs) in the cell’s cytoplasm (Figure 6C). In several MOGGCCM and T98G cells, swelling and rupture of the plasma membrane typical of necrosis were noticed (Figure 6D).

MTT assay and microscopic determination of cell death type induced by single sorafenib and LY294002 application allowed determination of the most efficient method for glioma cell elimination and the safest inhibitor concentration for normal cells. As such, 10 µM of PI3K and 1 µM of Raf inhibitor were used for further analysis.

### 3.3. Activity of Caspases

Caspases are well-defined apoptosis executioners, and they are divided into two major groups, initiators (such as 2, 8, 9, and 10) and effectors (3, 6, and 7). The first category determines the type of apoptosis pathway (extrinsic or intrinsic), the second is crucial for the unconditional induction of cell death by apoptosis. Therefore, analysis of 7, 8, and 9 caspases activity was performed. In MOGGCCM (Figure 7A) cells treated with LY294002, no significant increase in any caspase was observed, which was noticed in cells after single application of sorafenib. In the case of T98G (Figure 7B) sorafenib-treated cells, there was no significant growth in caspase activity compared to the control, whereas incubation of LY294002 caused an increase in caspases. A significant increase in caspase 7 activity was noticed in anaplastic astrocytoma and glioblastoma multiforme cells after combination of both inhibitors, and this was correlated with caspase 9 activity enlargement.

### 3.4. Bcl-2 and Beclin-1 Subcellular Localization Analysis

#### 3.4.1. Immunofluorescence

To investigate the localization of Bcl-2 and beclin-1 proteins and the role of their complex formation upon LY294002 or/and sorafenib activity, control and treated cells were fixed and then the immunofluorescence of proteins marked with AlexaFluor^TM^ was observed under a confocal microscope. To validate subcellular localization of Bcl-2 and beclin-1, nuclear stain Hoechst 33342 was used. As presented in Figure 8, significant changes in Bcl-2 and beclin-1 localization were noticed. In control cells (C), Bcl-2 was mainly observed near the mitochondria and beclin-1 in the cytoplasm, without visible correlation between them. In cells after PI3K inhibitor (L) incubation, where autophagy was the dominant type of death, both proteins were noticed in whole cells (also in nuclei); however, Bcl-2 was concentrated in the endoplasmic reticulum (ER), and it had no correlation with beclin-1. Simultaneous application of LY294002 and sorafenib (LS), where apoptosis was dominative, resulted in the formation of a characteristic apoptotic body and both proteins, Bcl-2 as well as beclin-1, were cumulated around them, and co-localization was observed.

#### 3.4.2. Single-Live-Cell Level Analysis

To characterize the physiological behavior of Bcl-2:beclin-1 complex in the context of apoptosis/autophagy switching in live cells, single-live-cell level experiments were performed. GFP-Bcl-2 and mCherry-beclin-1 were used, respectively. As was revealed in immunofluorescence assay in control (Figure 9, C) cells, the evaluated proteins were not co-localized. Bcl-2 was correlated with mitochondria, whereas beclin-1 was evenly dispersed throughout the cytoplasm, and neither were noticed in the nuclei. In LY294002-treated cells (Figure 9, L), apoptosis and autophagy regulators changed their localization and both were spotted in nuclei. Bcl-2 was concentrated in the ER and beclin-1 in cytoplasm. Non-significant co-localization was noted. The combination of PI3K and Raf inhibitors (Figure 9, LS) resulted in noticeable co-localization of Bcl-2 and beclin-1, and they were distributed near the apoptotic bodies formed.

### 3.5. Analysis of the Bcl-2:Beclin-1 Protein Complex through Immunoprecipitation Assay

It is known that direct binding of Bcl-2 and beclin-1 proteins matters in directing cells toward apoptosis or autophagy. To prove that the co-localization observed under the microscope was correlated with Bcl-2:beclin-1 complex formation, an immunoprecipitation assay was performed (Figure 10).

As presented in Table 1, immunoprecipitation revealed that the complex of both proteins in the MOGGCCM and T98G lines was not detected in the control cells. Sorafenib increased this correlation in anaplastic astrocytoma, despite the low level of apoptosis (5%), which was not noticed in glioblastoma multiforme (autophagy was dominant). Interestingly, in T98G, LY294002-treated cell complex was not observed, despite the domination of apoptosis (autophagy was also at a significant level). LY294002 and sorafenib in combination had the highest influence on the correlation of the studied proteins in both cancer cell lines, which was correlated with increased level of apoptosis.

### 3.6. Involvement of Bcl-2 and Beclin-1 Expression in Type of Cell Death Induction after LY294002 and Sorafenib Treatment

To receive direct proof of the involvement of Bcl-2 and beclin-1 in cell death induction upon LY294002 and sorafenib application, specific siRNAs were used. Transfection efficiency was determined using control siRNA, as presented in Figure 11E,F. Expression silencing success was verified through immunoblotting assay, which revealed that siRNA-Bcl-2 and siRNA-beclin-1 were effective in decreasing the protein expression (Figure 11G,H). Analysis of the type and level of cell death in the transfected cells revealed that apoptosis was dominant type of death, despite blocking the expression of beclin-1, both in MOGGCCM (Figure 11A,C) and T98G cells (Figure 11B,D); however, after blocking Bcl-2 expression, it was higher. The combination of LY294002 and sorafenib was the most effective in sensitizing Bcl-2-silenced anaplastic astrocytoma and glioblastoma multiforme cells to apoptosis induction. An analogous situation was observed in beclin-1-silenced T98G cells, whereas in MOGGCCM, single PI3K inhibitor usage was slightly more effective than simultaneous application. No necrotic effects were noticed in any of the tested variants.

## 4. Discussion

Anaplastic astrocytoma (AA) and glioblastoma multiforme (GBM) are the most malignant and aggressive tumors of the central nervous system in adults. The prognosis for patients diagnosed with AA or GBM is very poor, and they are incurable at present. The average survival rate ranges from 9 months to 5 years, depending on the degree of malignancy and the age of the patient. The malignant nature and high mortality of these tumors correspond with high proliferation and migration rates and an aggressive growth pattern. Their resistance to standard treatment is correlated with programmed cell death (PCD) elusion, due to overexpression of intracellular survival pathways PI3K-Akt/PKB-mTOR and Ras-Raf-MEK-ERK [9,10]. In our previous studies, we discovered that these pathways play a key role in the drug resistance of glioma cells. Blocking the survival signal transduction using their specific inhibitors, LY294002 (PI3K inhibitor) and sorafenib (Raf inhibitor), led to sensitizing AA and GBM cells to apoptosis induction after Temozolomide treatment [2,3]. A combination of these inhibitors was more efficient than a single application for glioma cell elimination. However, not only the cytotoxic effect is important but also the type of cell death. Necrosis is a very intrusive mechanism of cellular death in cancers, in view of its character and consequences, which lead to highly inflammatory conditions, which is why cancers are called an inflammatory disease, so this type of death is highly undesirable [11]. On the other hand, autophagy is an ambiguous and controversial mechanism in PCD, and it is believed this may lead to cancer cell survival and increased drug resistance [12,13]. In the light of these factors, apoptosis seems to be the most preferable process of glioma elimination, which is why modern strategies based on its induction are needed more than the other types of death. As stated in the literature, apoptosis and autophagy are dynamic, switchable processes in cells. Their molecular switch may be a complex of two proteins: the anti-apoptotic Bcl-2 and the pro-autophagic beclin-1 [4,5]. The beclin-1 protein is a key marker of autophagy [14], which combined with Bcl-2 protein, inhibits the formation of the pre-autophagosomal structure and thus inhibits autophagy and increases apoptosis induction. Therefore, apoptosis is inhibited when these proteins are in a free state, and the autophagy process can proceed unhindered. Considering the fact that autophagy is increasingly less desirable than apoptosis in the context of eliminating cancer cells, such a specific switch may be a molecular target in the development of new, more effective therapies.

Therefore, the aim of the present study was to investigate, for the first time, the role of Bcl-2:beclin-1 in “switching” between apoptosis and autophagy in glioma cells using the combined action of LY294002 and sorafenib.

Many studies have also presented the cytotoxic potential of PI3K (phosphatidylinositol 3-kinase) and Raf inhibitor. LY294002 is a PI3K inhibitor, and its cytotoxicity was tested on IA breast cancer (MCF-7) [15], metastatic melanoma [16], oral squamous cell carcinoma (SCC-25) [17], and esophageal cancer (EC 9706) [18], and it resulted in cell viability decreasing. Our results correspond with those data, and PI3K inhibitor reduced there viability of anaplastic astrocytoma (MOGGCCM) and (T98G) in all tested concentrations. Comparable results were observed in sorafenib-treated cancer cells. Sorafenib is an inhibitor of several kinases that are involved in cancer cell proliferation and tumor angiogenesis, including Raf. Its cytotoxic effect was also evaluated in different cell lines, such as hepatoma (PLC/PRF/5) [19], hepatocellular carcinoma (Hep G2) [20], thyroid cancer (FTC133) [21], acute myeloid leukemia (HL-60, MOLM-13, HEL, and OCI-AML2) [22], and chronic myelogenous leukemia in blast phase (K-562) [23]. Decrease in the viability of these cells was also observed in our results from MOGGCCM and T98G sorafenib-treated cells. However, different concentrations of survival pathway inhibitors may not only lead to cancer cell elimination but could also affect normal cells. That is why the MTT test was also performed on two normal, astrocyte (NHA) and oligodendrocyte (OLN-93), cell lines, for determination of the most effective drug concentration. Low doses of LY294002 and sorafenib did not significantly affect normal cell viability, which suggested that PI3K and Raf inhibitor usage may be safe. Despite these promising results, not only the cytotoxic effect is crucial in cancer cell elimination with no harmful effects on normal cells but also the type of death. In our current and previous studies [2,3], we showed that, in MOGGCCM LY294002-treated cells, autophagy was mainly observed, as well as in T98G sorafenib-treated cells, and we revealed that their combination was more efficient in glioma cell elimination through apoptosis induction. The PI3K and Raf inhibitor concentrations determined for further analysis were not affected in any of PCD-type inductions in normal cells. What is more important, no significant necrosis was observed in the normal or cancer cells. The types of cell death were also confirmed using characteristic changes in cellular morphology at ultrastructural level with transmission electron microscopy (TEM). In case of apoptosis, typical apoptotic bodies were noticed, while autophagic cells were identify using the presence of acidic vesicular organelles within cytoplasm. Apoptosis induction may be performed using two pathways, extrinsicly connected to death-toll receptors (DR) and intrinsic, which is associated with cytochrome C release from mitochondria, as well as a perforin/granzyme pathway. DR and mitochondrial paths are strictly regulated by caspase cascade; for example, caspase 3, 7 (effectors), 8 and 9 (initiators). Appropriate activation of one of these results in an external or internal form of apoptosis activation. Since our previous studies about LY294002 and sorafenib proapoptotic action [3] revealed an inconclusive increase in caspase 3 activation, which is crucial in apoptosis induction, a caspase 7, 8, and 9 activity assays were performed. It is known that apoptotic death may also be initiated in cell via caspase 7. The results obtained showed an increase in caspase 7 activity in sorafenib-treated anaplastic astrocytoma cells, as well as after combination with LY294002, which was correlated with a boost in caspase 9 activity. In glioblastoma multiforme cells, similar consequences were observed after single PI3K inhibitor application and with Raf inhibitor combination. This seems to not be a common observation, because an increase in caspase 3 activity after LY294002 treatment models was mainly observed in the different research so far [24,25]. However, there is research corresponding with our results, where 3 and 7 caspase activity improvements were observed in retinal pigment epithelium cells after PI3K inhibitor treatment [26]. Increases in caspase 9 activity suggested that apoptosis proceeded via an intrinsic, mitochondrial pathway.

As is known, programmed cell death mechanisms are coexisting processes, and they are interdependent and mutually influenced. This is possible due to the interactions between their key regulators, the anti-apoptotic protein Bcl-2 and the specific autophagy marker beclin-1 [27]. The created complex is known as a molecular “switch” between apoptosis and autophagy. It has been previously observed that a significant increase in the level of apoptosis in glioma cells is closely correlated with the presence of a complex of these proteins. The formation of the Bcl-2:beclin-1 complex also depends on the intracellular localization of the proteins [4]. Therefore, after conscientious investigation of the level and type of cell death after single and combined action of LY294002 and sorafenib in anaplastic astrocytoma and glioblastoma multiforme cells, localization changes and co-localization of Bcl-2 and beclin-1 proteins were analyzed. Immunofluorescence assay (IF) revealed significant changes in PCD regulator intracellular localization. In MOGGCCM and T98G cells after PI3K or Raf inhibitor application, Bcl-2 and beclin-1 were located within the whole cell and in nuclei, which was not observed in control cells. Moreover, Bcl-2 accumulation was noticed in the endoplasmic reticulum (ER) after single drug usage, whereas in control cells, it was located near mitochondria. Simultaneous application of LY294002 and sorafenib in anaplastic astrocytoma, as well as in glioblastoma multiforme, affected the co-localization of Bcl-2 and beclin-1, which were distributed around the formed apoptotic bodies. Some studies showed that the nuclear localization of beclin-1 may lead to autophagy activation due to starvation or stressful conditions [28]. Moreover, beclin-1 may interact with PI3K, which can lead to PtdIns(3)P monophosphate in the trans-Golgi network and plays a key role in autophagosome formation [29,30]. This partly explains the correlation in our results between PI3K expression inhibition and changes in autophagy level correlated with beclin-1 nuclear orientation. Similar observations were noticed after Raf inhibitor usage; however, exact correlation between Raf kinase and beclin-1 is not known and more studies are needed. On the other hand, Bcl-2 subcellular localization is also crucial in the apoptosis/autophagy crosstalk. Endoplasmic reticulum attached Bcl-2 is able to inhibit autophagy and promote the apoptosis process, while mitochondrial Bcl-2 has an anti-apoptotic effect by releasing beclin-1, which in turn promotes the autophagy process [31,32]. The above observations were made in fixed cells, which is why the analysis of physiological behavior of Bcl-2:beclin-1 complex in the context of apoptosis/autophagy “switching” in vivo confocal microscopy experiments was performed. The single-live-cell level observations showed equivalent results for Bcl-2 and beclin-1 localization changes after single LY294002 or sorafenib incubation in MOGGCCM and T98G cells and their co-localization after combined inhibitors treatment. To gain additional evidence for the involvement of Bcl-2 and beclin-1 complex formation upon LY294002 and sorafenib treatment, specific siRNAs were used to block their expression, and the dominant type of death and its level was analyzed. The used siRNAs were effective in Blc-2 and beclin-1 expression silencing. In both MOGGCCM and T98G treated cell lines after silencing of Bcl-2 expression, apoptosis was the dominant type of death in all variants, and the highest level was observed in cells after combined inhibitor action, which was suspected; however, interestingly, no apoptosis or autophagy was observed in control cells. An unexpected result was the domination of apoptosis after silencing of beclin-1 expression. It seemed that the anti-apoptotic activity of Bcl-2 should be increased after prevention of complex with beclin-1; however, the only apoptosis was observed in treated cells. This may confirm the dual role of Bcl-2 in different types of programmed cell death induction.

Expanding knowledge of the central nervous system tumors, such as malignant gliomas, based on the role of the molecular switch, the Bcl-2:beclin-1 complex may be the basis of modern anticancer therapy development. In particular, in the context of PI3K-Akt/PKB-mTOR and Ras-Raf-MEK-ERK, inhibition by LY294002 and sorafenib, and their combination, during cancer treatment would be beneficial.

## 5. Conclusions

The results of this research showed that the usage of the intracellular survival signaling pathway inhibitors PI3K-Akt/PKB-mTOR and Ras-Raf-MEK-ERK in simultaneous application resulted in the formation of Bcl-2 and beclin-1 complexes, which was correlated with induction of apoptosis in glioma cells. A better understanding of the exact mechanisms of the “switching” processes between apoptosis and autophagy through PI3K and Raf inhibitor usage in the context of the Bcl-2:beclin-1 complex formation opens up new perspectives in the fight against gliomas, both in the context of designing new therapies and exploring their resistance to elimination through programmed death.

The obtained results will significantly expand the current knowledge on the biology of the central nervous system tumors and may be a basis for new perspectives in modern therapy development.

## Figures and Tables

**Figure 1 cells-12-02670-f001:**
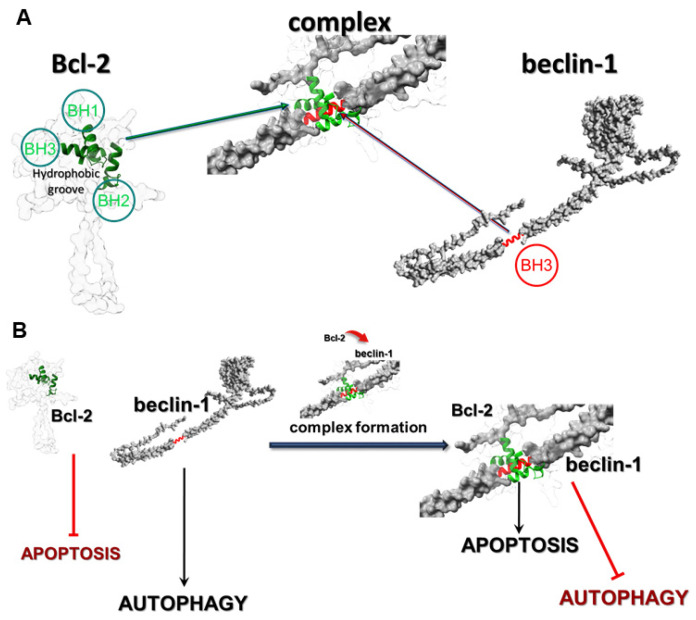
The process of Bcl-2:beclin-1 complex formation (**A**) and its function for switching between apoptosis and autophagy (**B**). UniProt IDs for created conformations: Bcl-2—AF-P10415-F1, beclin-1—AF-Q6ZNE5-F1 [8].

**Figure 2 cells-12-02670-f002:**
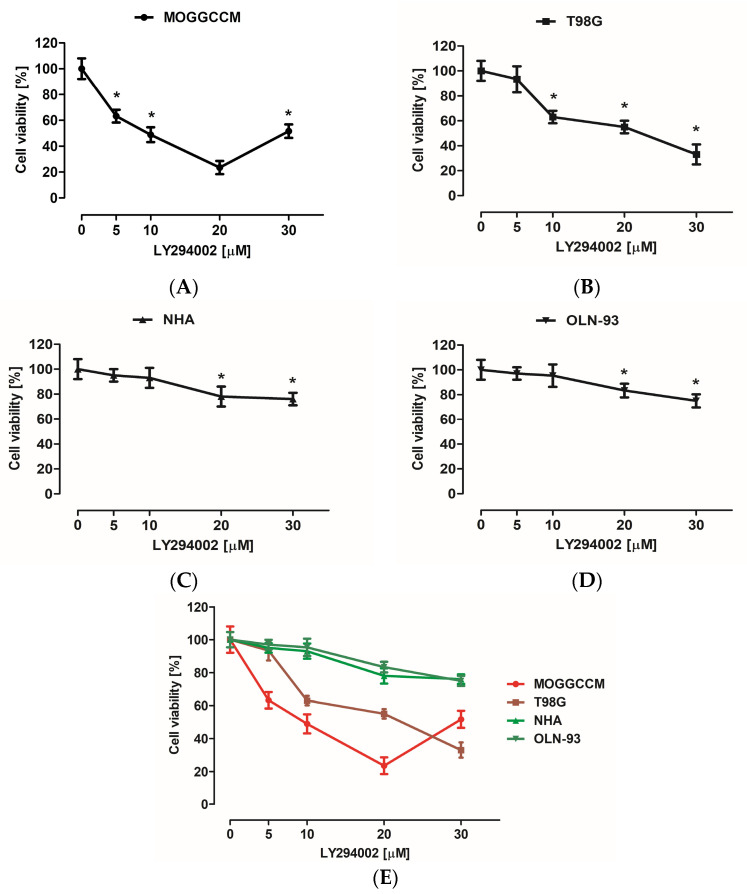
Cytotoxic effect of different concentrations of LY294002 (PI3K inhibitor) on anaplastic astrocytoma (**A**), glioblastoma multiforme (**B**), astrocytes (**C**), and oligodendrocytes (**D**); (**E**) graph as a comparison; measured by means of MTT assay in vitro. Results are expressed as the mean of control (0 µM) of 6 independent experiments; * *p* < 0.05 compared to control.

**Figure 3 cells-12-02670-f003:**
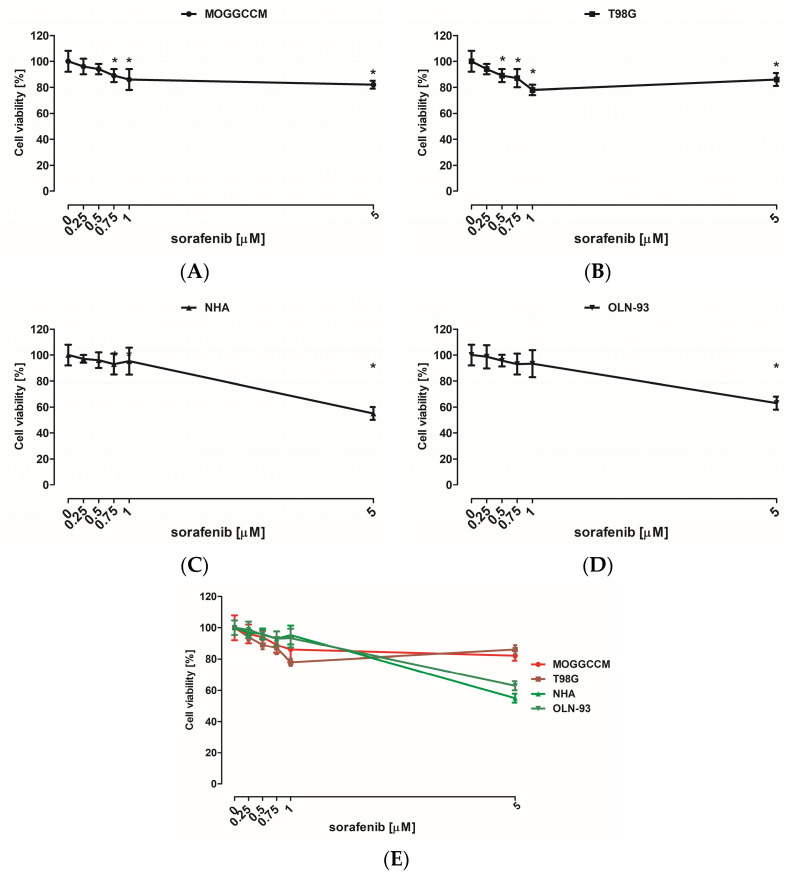
Cytotoxic effect of different concentrations of sorafenib (Raf inhibitor) on anaplastic astrocytoma (**A**), glioblastoma multiforme (**B**), astrocytes (**C**), and oligodendrocytes (**D**); (**E**) graph as a comparison; measured by means of MTT assay in vitro. Results are expressed as the mean of control (0 µM) of 6 independent experiments; * *p* < 0.05 compared to control.

**Figure 4 cells-12-02670-f004:**
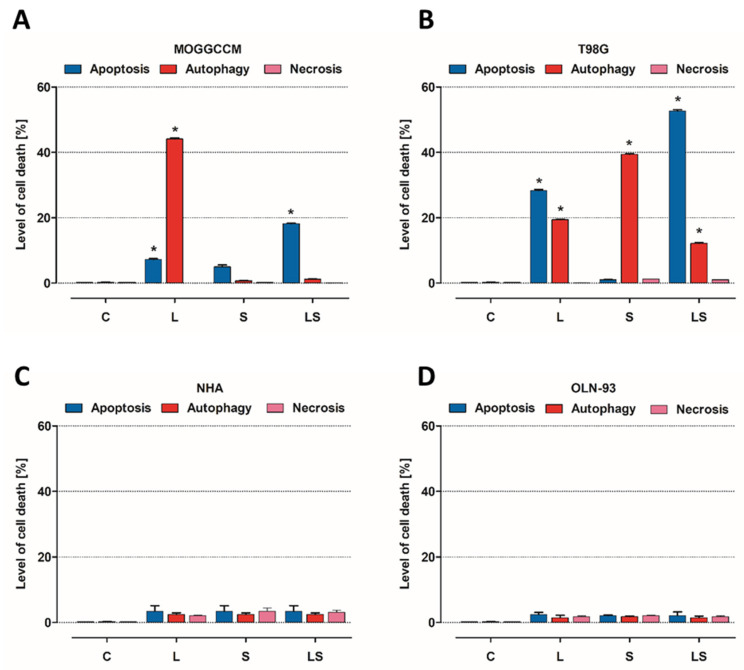
Types and level of cell death observed in anaplastic astrocytoma (**A**), glioblastoma multiforme (**B**), astrocytes (**C**), and oligodendrocytes (**D**) after LY294002 (L) and sorafenib (S) in single and simultaneous (LS) application; * *p* < 0.05 compared to control.

**Figure 5 cells-12-02670-f005:**
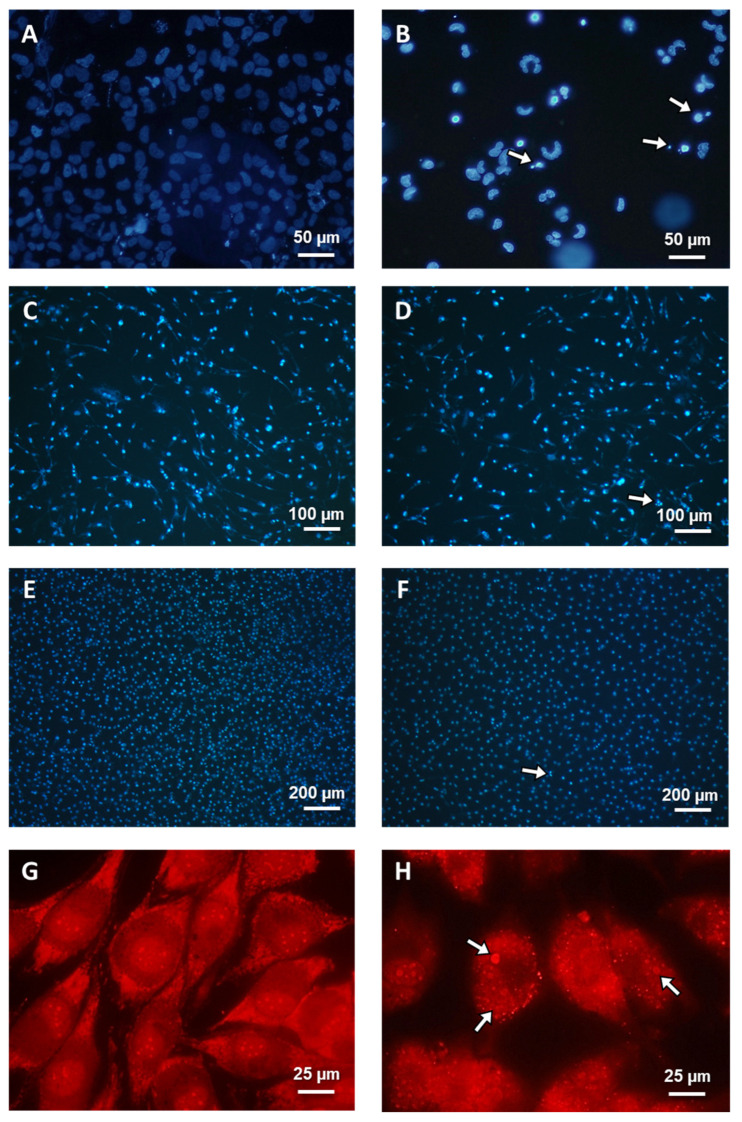
Representative photos of microscopic analysis of characteristic morphology in Hoechst 33342 and propidium iodide T98G (**A**,**B**) NHA (**C**,**D**) and OLN-93 (**E**,**F**) stained cells and in acridine orange T98G dyed cells (**G**,**H**). White arrows—apoptotic (**B**,**D**,**F**) and autophagic (**H**) cells. Zoomed (**D**,**F**) pictures are attached as a Appendix A.

**Figure 6 cells-12-02670-f006:**
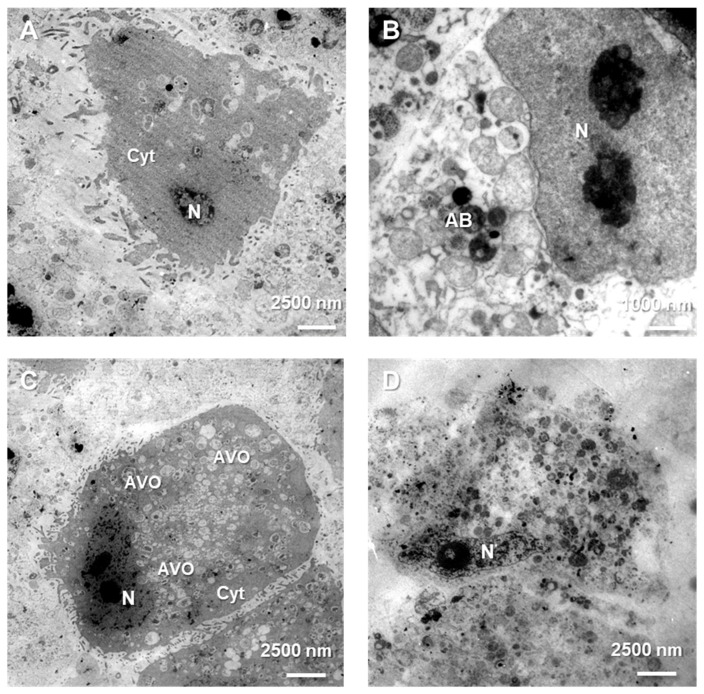
Representative photos of ultrastructural observations of MOGGCCM and T98G cells in TEM microscopy. Characteristic morphology of normal (**A**), apoptotic (**B**), autophagic (**C**), and necrotic cells (**D**). N—nucleus, Cyt—cytoplasm, AB—apoptotic bodies, AVO—acidic vesicular organelles.

**Figure 7 cells-12-02670-f007:**
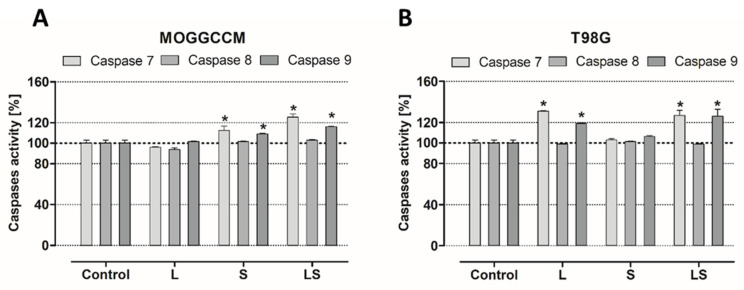
Analysis of 7, 8, and 9 caspases in MOGGCCM (**A**) and T98G (**B**) cells after LY294002 (L) or sorafenib (S) treatment. LS—combination of drugs; * *p* < 0.05 compared to control.

**Figure 8 cells-12-02670-f008:**
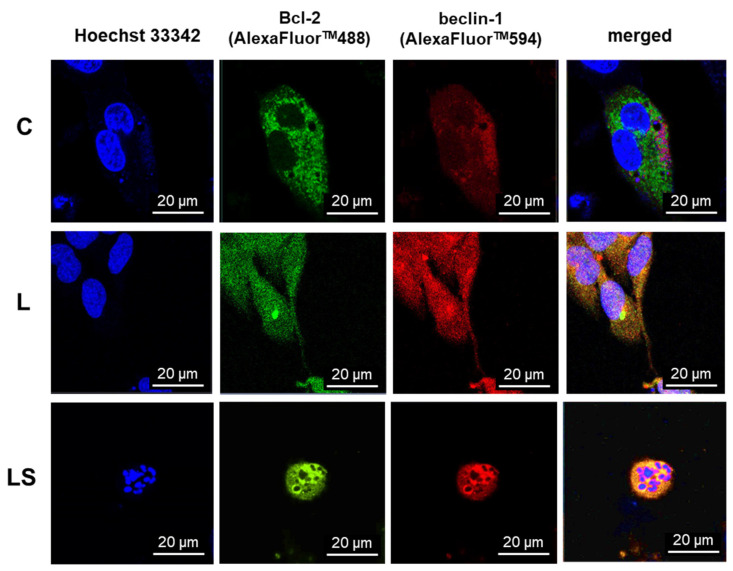
Representative photos of the localization and co-localization of Blc-2 and beclin-1 proteins marked by AlexaFluor^TM^ antibodies in T98G control (C) and treated (L, LS) cells. Nuclear Hoechst 33342 staining performed. C—untreated cells, L—LY294002, LS—LY294002 + sorafenib.

**Figure 9 cells-12-02670-f009:**
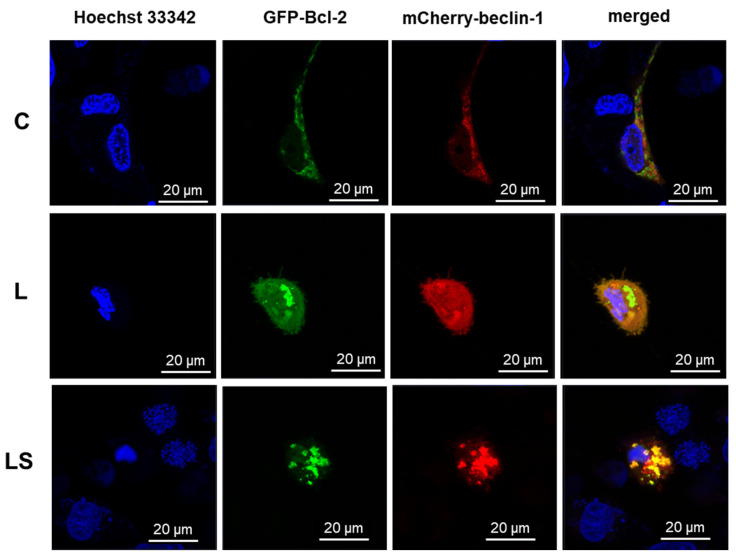
Representative photos of localization and co-localization of Blc-2 and beclin-1 proteins fused with fluorescent proteins GFP and mCherry in T98G control (C) and treated (L, LS) cells. Nuclear Hoechst 33342 staining performed. C—control, L—LY294002, LS—LY294002 + sorafenib.

**Figure 10 cells-12-02670-f010:**
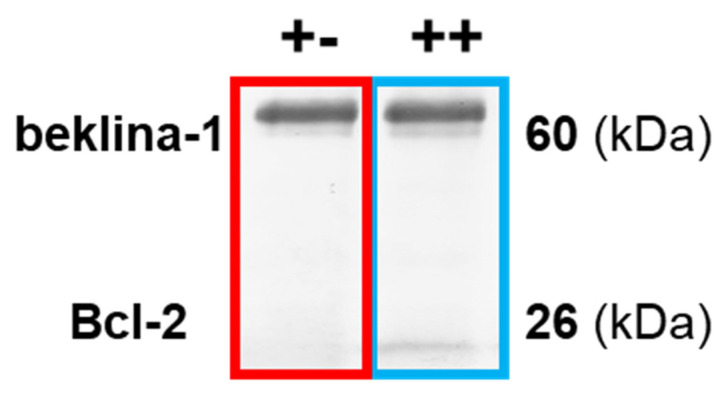
Representative fragment of blot presented Bcl-2:beclin-1 complex formation, after co-immunoprecipitation assay. +- not presence, ++ complex presence. Whole blot membranes are attached in the Appendix A.

**Figure 11 cells-12-02670-f011:**
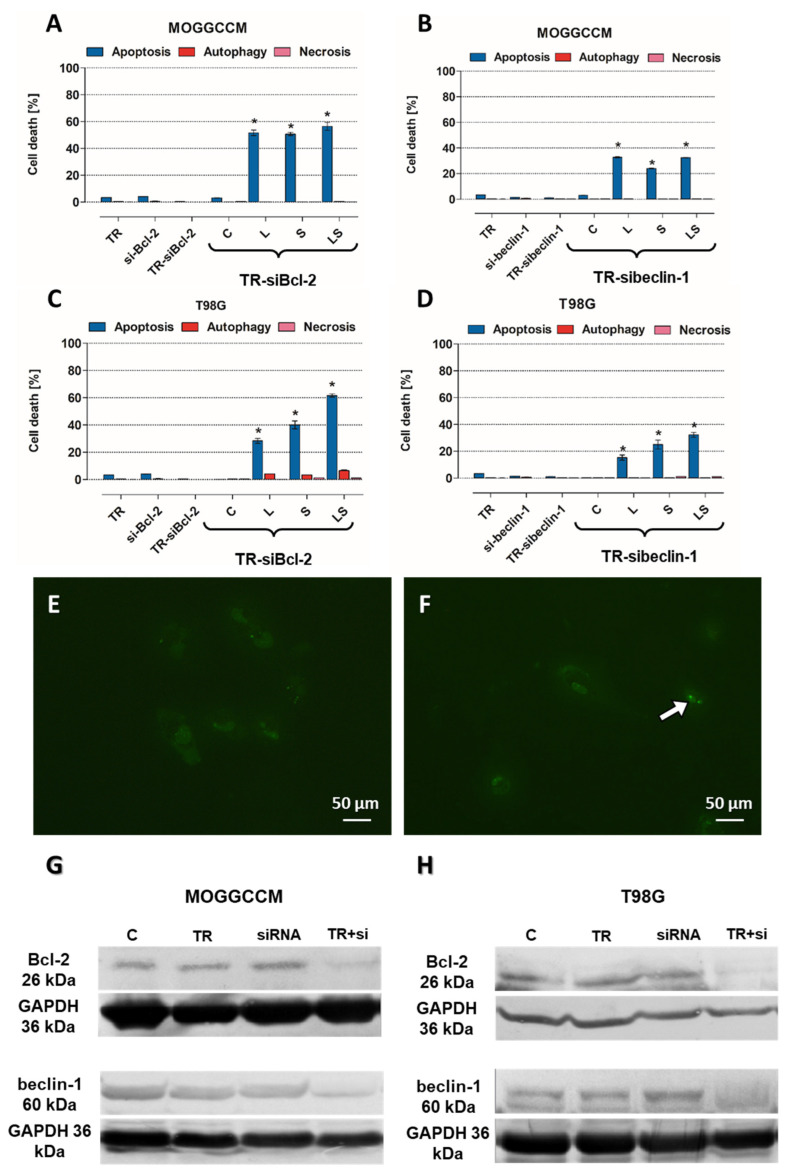
Results of blocking the expression of Bcl-2 and beclin-1 proteins in MOGGCCM and T98G control (C) and treated cells (**A**–**D**). (**E**)—non-transfected cells, (**F**)—control siRNA transfected cells (white arrow); (**G**,**H**)—expression silencing efficiency on immunoblot membranes; L—LY294002, S—sorafenib, LS—LY294002 + sorafenib; * *p* < 0.05 compared to control. Whole blot membranes attached as Appendix A.

**Table 1 cells-12-02670-t001:** Influence of LY294002 (L) or sorafenib (S) in single and simultaneous (LS) application on Bcl-2:beclin-1 complex formation in MOGGCCM and T98G cells. ++ complex presence, +- not presence.

MOGGCCM				
	beclin-1	Bcl-2	apoptosis	autophagy
C	+	-	-	-
L	+	-	7%	**45%** *
S	+	+	5%	1%
LS	+	+	**19%** *	-
T98G				
	beclin-1	Bcl-2	apoptosis	autophagy
C	+	-	-	-
L	+	-	28% *	20%
S	+	-	5%	**40%** *
LS	+	+	**52%** *	12%

●—apoptosis dominant, ●—autophagy dominant. * *p* < 0.05 compared to control.

## Data Availability

The data presented in this study are available from the corresponding author on request.

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
