# Peer review of "The Role of Bcl-2 and Beclin-1 Complex in “Switching” between Apoptosis and Autophagy in Human Glioma Cells upon LY294002 and Sorafenib Treatment"

_cells, 2023, doi:10.3390/cells12232670_

Round 1

Reviewer 1 Report

Comments and Suggestions for Authors

Dear Authors,

Your manuscript on the role of Bcl-2 and beclin-1 complex in human glioma cells is a valuable contribution to the field of glioma research. All experiments are very well described and interpretation is very clear discussed. Nevertheless, I do have some points that should be considered:

1.       The cell line MOGGCCM you use is a rarely used cell line and therefore not very well characterized. Based on the current WHO guidelines this cell line is most probably not an anaplastic astrocytoma cell line. Cell lines with IDH mutation are very rare and not easy to cultivate. Therefore, I recommend characterizing this cell line a bit further. Checking for IDH mutation would be my first choice. Should there be no mutation in IDH1 or IDH2, the cell line has to be considered a glioblastoma cell line and the manuscript accordingly adapted to this. A more accurate characterization of this cell line would also help interpreting the results. T98G cell line for example is PTEN mutant and MOGGCCM probably not. This may influence and partially explain the differences in drug response. Some genetic data of both cell lines can be found on COSMIC cell line project for example.

2.       A discrepancy between the different methods used has to be discussed in more detail. In line 336, 457 and later in the discussion (line 526) it is mentioned that there is no necrosis, but in the EM analysis in line 355-356 it is stated that signs of necrosis are frequent. Please explain these differences.

Minor remarks

·         Carefully check for spelling errors (lines 133, 148, 191…)

·         The writing of “degree” C is in some cases a bit odd. It is in the middle of the line and not on top right corner of the “C” (lines  146, 219 and 234)

·         In Figure 4 it is not very clear where the significance is referring to.  

·         Figure 5 C-F is not very informative because the structures of interest are depicted to small. I would recommend to show all cells in the same magnification.

Best regards

Reviewer 2 Report

Comments and Suggestions for Authors

This paper tried to investigate the role of Bcl-2 and beclin-1 complex in “switching” between apoptosis and autophagy upon combined treatment with PI3K inhibitor LY294002 and Raf inhibitor sorafenib. Using a series of experiments on a glioblastoma cell line and an anaplastic astrocytoma cell line, the authors checked the effect of the proposed drug combination on cell growth inhibition, apoptosis and autophagy. The effect of the drugs was observed on caspase activity, Bcl-2 and beclin-1 subcellular localization and their protein complex precipitation.

While the question pursued here is of importance, the main issue in this study is lack of novelty that is presentation of results from previous study. Namely, results showing the effect of LY294002 on glioma growth inhibition, types and level of cell death observed in glioma cells upon treatment with LY294002 and sorafenib and their combination as well as their effect on activity of caspase 8 and 9 are presented in already published manuscripts.

Novel results in this study are confirmation of formation of Bcl-2 and beclin-1 complex and their role in apoptosis induction. However, Bcl-2 and beclin-1 subcellular localization analysis has major limitation. How can authors be sure that Bcl-2 is correlated with the mitochondria in control cells and concentrated in ER upon treatment with LY294002 without using specified marker for these organelles? Furthermore, if you want to evaluate whether Bcl-2 co-localizes with beclin-1 after combined treatment with LY294002 and sorafenib you need to perform co-localization analysis using the Pearson correlation coefficient that measures the degree of correlation between Bcl-2 and beclin-1.  Namely, Pearson correlation coefficient measures the strength of a linear relationship between fluorescent intensities from the two images and produces values ranging from 1 (perfect positive correlation) to −1 (perfect inverse correlation), with 0 representing a random distribution.

Comments on the Quality of English Language

Minor editing of English language is required. 

Round 2

Reviewer 2 Report

Comments and Suggestions for Authors

The authors answered and explained all my concerns and I recommend this manuscript for publication in Cells.